# α5-GABAA Receptor Modulation Reverses Behavioral and Neurophysiological Correlates of Psychosis in Rats with Ventral Hippocampal Alzheimer’s Disease-like Pathology

**DOI:** 10.3390/ijms241411788

**Published:** 2023-07-22

**Authors:** Nicole E. Eassa, Stephanie M. Perez, Angela M. Boley, Hannah B. Elam, Dishary Sharmin, James M. Cook, Daniel J. Lodge

**Affiliations:** 1Department of Pharmacology and Center for Biomedical Neuroscience, UT Health San Antonio, San Antonio, TX 78229, USA; eassa@livemail.uthscsa.edu (N.E.E.); boley@uthscsa.edu (A.M.B.); elamh@livemail.uthscsa.edu (H.B.E.); lodged@uthscsa.edu (D.J.L.); 2South Texas Veterans Health Care System, Audie L. Murphy Division, San Antonio, TX 78229, USA; 3Department of Chemistry and Biochemistry, Milwaukee Institute of Drug Discovery, University of Wisconsin-Milwaukee, Milwaukee, WI 53211, USA; dsharmin@uwm.edu (D.S.); capncook@uwm.edu (J.M.C.)

**Keywords:** Alzheimer’s disease, psychosis, dopamine, ventral hippocampus, parvalbumin, α5-GABAA

## Abstract

Of the 35 million people in the world suffering from Alzheimer’s Disease (AD), up to half experience comorbid psychosis. Antipsychotics, used to treat psychosis, are contraindicated in elderly patients because they increase the risk of premature death. Reports indicate that the hippocampus is hyperactive in patients with psychosis and those with AD. Preclinical studies have demonstrated that the ventral hippocampus (vHipp) can regulate dopamine system function, which is thought to underlie symptoms of psychosis. A viral-mediated approach was used to express mutated human genes known to contribute to AD pathology: the Swedish (K670N, M671L), Florida (I716V), and London (V717I) mutations of amyloid precursor protein and two mutations (M146L and L286V) of presenilin 1 specifically in the vHipp, to investigate the selective contribution of AD-like pathology in this region. We observed a significant increase in dopamine neuron population activity and behavioral deficits in this AD-AAV model that mimics observations in rodent models with psychosis-like symptomatologies. Further, systemic administration of MP-III-022 (α5-GABAA receptor selective positive allosteric modulator) was able to reverse aberrant dopamine system function in AD-AAV rats. This study provides evidence for the development of drugs that target α5-GABAA receptors for patients with AD and comorbid psychosis.

## 1. Introduction

Alzheimer’s Disease (AD) is the most common neurodegenerative disease, affecting an estimated 6.7 million Americans age 65 and older [1] and is known to cause a progressive cognitive decline, but it is less appreciated that half of these patients also suffer with comorbid psychosis (hallucinations and delusions) [2]. Elderly patients are unable to be treated with antipsychotics (the standard-of-care) given that the Food and Drug Administration issued a black box warning contraindicating their use in the elderly due to an increased risk of premature death. AD patients suffering from comorbid psychosis are clearly in need of novel therapeutic options.

Dopamine system dysfunction has been demonstrated to underlie symptoms of psychosis and is likely attributable to aberrant regulation by afferent brain regions [3,4]. The hippocampus is one such region upstream of the dopamine system that is a key site of pathology in patients with AD [5] and displays aberrant activity in psychosis [6]. In preclinical studies, our laboratory, along with others, has shown that hyperactivity of the ventral hippocampus (vHipp) drives dopamine system dysfunction in the ventral tegmental area (VTA) via a multi-synaptic circuit in rodent models used to study psychosis [7]. Further, we have shown that increasing expression of the α5 subunit of the GABAA receptor in the vHipp reverses neuronal and behavioral deficits analogous to psychosis [8]. Given that the endogenous expression of α5 is highly specific to the hippocampus [9], systemic administration of positive allosteric modulators of α5 (α5-PAMs), which were developed as nootropic drugs that work by normalizing the excitation/inhibition (E/I) balance, may serve as a novel therapeutic option for patients with comorbid psychosis by reversing aberrant hippocampal drive of the dopamine system.

Previous studies have demonstrated that amyloid beta (Aβ) plaques, which accumulate in the hippocampus and other cortical areas, induce hyperactivity in the vHipp and psychosis-like neuronal and behavioral pathology in a rat model used to study sporadic AD [10]. In order to investigate this pathophysiology in a circuit-specific manner, and determine whether an α 5-PAM, MP-III-022, is able to reverse this, here we virally expressed two mutated human genes known to contribute to AD pathology, the Swedish (K670N, M671L), Florida (I716V), and London (V717I) mutations of human amyloid precursor protein (hAPP) and two mutations (M146L and L286V) of presenilin 1 (hPSEN1), specifically in the vHipp. Once the rats reached adulthood, corresponding to the age of AD pathology in humans, we examined neuronal pathology in the vHipp and VTA and cognitive and behavioral phenotypes consistent with AD and comorbid psychosis. These studies establish a circuit-specific rationale for targeting the hippocampus, using the selective α5-PAM, MP-III-022 [11], to reverse downstream dopamine system dysfunction, which will lay the groundwork for developing therapeutic options for AD patients with comorbid psychosis.

## 2. Results

### 2.1. AAV-AD Rats Display Impairments in Hippocampal Activity That Are Reversed by Systemic Administration of the Selective α5-PAM, MP-III-022

Hippocampal hyperactivity has been observed in rodent models that display psychosis-related pathologies [12,13,14]; however, AAV-AD rats (n = 77 neurons; 0.78 ± 0.07 Hz) did not exhibit a vHipp firing frequency different from control rats (n = 93 neurons; 0.73 ± 0.06 Hz; Figure 1B). Additionally, vHipp firing rates of control (n = 87 neurons; 0.87 ± 0.06 Hz) or AAV-AD rats (n = 78 neurons; 0.70 ± 0.05 Hz) were not affected by the selective α5-PAM, MP-III-022 (Figure 1B). Recordings of spontaneous oscillatory activity in the vHipp (Figure 1C–F) revealed a main effect (n = 4–7 rats per group; three-way ANOVA; F_Strain_(1, 119) = 0.32; F_Frequency_(4, 119) = 19.31; *p* < 0.001; F_Treatment_(1, 119) = 2.58; F_Strain×Treatment_(1, 119) = 6.95; *p* = 0.01; Holm Sidak; *t* = 2.51; *p* = 0.01), whereby we observed an overall decrease in oscillatory activity in AAV-AD rats when compared to controls. Interestingly, this was reversed by the systemic administration of MP-III-022 (*t* = 2.83; *p* = 0.006; Figure 1C,D).

### 2.2. AAV-AD Rats Exhibit Aberrant Dopamine System Function That Is Reversed by Systemic Administration of the Selective α5-PAM, MP-III-022

Various models used to study psychosis display a significant increase in VTA dopamine neuron population activity [7,14,15,16,17]. Consistent with these data, the population activity of AAV-AD rats (n = 10; 1.60 ± 0.13 cells per track) was significantly higher than controls (n = 10 rats; 1.05 ± 0.08 cells per track; two-way ANOVA; F_Strain_(1, 34) = 6.61; *p* = 0.02; F_Treatment_(1, 34) = 9.48; *p* = 0.004; F_Strain×Treatment_(2, 34) = 8.25; *p* = 0.007; Holm-Sidak; *t* = 4.16; *p* < 0.001; Figure 2B). Systemic administration of the selective α5-PAM, MP-III-022 reversed increases in population activity in AAV-AD rats (n = 8 rats; 1.00 ± 0.06 cells per track; *t* = 4.29; *p* < 0.001) and did not have an effect in control rats (n = 7 rats; 1.03 ± 0.08 cells per track). No significant differences in average firing rate (n = 44–98 dopamine neurons per group; Figure 2C) or bursting pattern (n = 44–98 dopamine neurons per group; Figure 2D) were observed between any groups.

### 2.3. AAV-AD Rats Do Not Display Cognitive Deficits but Exhibit Specific Deficits in a dopamine-Dependent Behavior, Which Was Reversed By Systemic Administration of the Selective α5-PAM, MP-III-022

Deficits in cognitive tasks are a consistent observation in rodent models of AD; however, given that this is a vHipp-specific model, no deficits in spatial learning and memory were observed. Here, we calculated the percent spontaneous alternations in the Y-maze and did not observe any difference between groups (n = 9–11 rats per group; Figure 3A), indicating that this model does not produce working memory deficits. Similarly, sensorimotor gating deficits have been observed in models used to study psychosis, as well as in models of AD; however, we did not observe a PPI deficit between groups (n = 10–11 rats per group; Figure 3B). Rodent models used to study psychosis commonly display deficits in dopamine-dependent measures, such as exaggerated responses to acute systemic administration of psychotomimetic drugs. Here, we demonstrate that baseline locomotor activity is consistent across all groups (n = 10–11 rats per group), while MK-801-induced locomotor activity was elevated in AAV-AD rats when compared to controls (three-way ANOVA; F_Strain_(1, 385) = 0.008; F_Treatment_(1, 385) = 0.67; F_Time_(8, 385) = 3.35; *p* = 0.001; F_Strain×Treatment_(1, 385) = 10.85; *p* = 0.001; Holm Sidak; *t* = 2.23; *p* = 0.026; Figure 3C). Further, this increase was reversed following systemic administration of the selective α5-PAM, MP-III-022 (Holm-Sidak; *t* = 2.87; *p* = 0.004).

### 2.4. Parvalbumin Positive Interneurons are Decreased in AAV-AD Rats

A decrease in the number of PV interneurons has been previously reported in postmortem studies from AD patients [18,19] and rodent models used to study psychosis [13,20]. Further, this has been correlated with VTA dopamine neuron population activity [15]. Here, we found a significant decrease in the number of PV neurons in the vHipp of AAV-AD rats (n = 13 rats; 5.79 ± 0.62 PV cells/mm^2^) when compared to controls (n = 13 rats; 7.82 ± 0.64 PV cells/mm^2^; *t*-test; *t* = 2.28 with 24 degrees of freedom; *p* = 0.03; Figure 4A). In addition, there is a trend whereby a decrease in vHipp PV interneurons corresponds with an increase in VTA dopamine neuron population activity (n = 13; linear regression; R = 0.40; r^2^ = 0.16; analysis of variance; F_Regression_(1, 12) = 2.10; *p* = 0.18; Figure 4C).

## 3. Discussion

Every year, in the United States alone, AD accrues a healthcare system burden of $340 billion [21]. Half of these AD patients also suffer from comorbid psychosis [2], which disproportionately contributes to the burden of the illness on society due to more advanced disease stage and increased adverse health outcomes [22]. Among these is an increased risk for mortality, and drugs that directly target dopamine system dysfunction that underlies symptoms of psychosis further increase the risk of death two-fold in the elderly population [23]. There is clearly a need to develop therapeutics that alleviate symptoms of psychosis in this patient population.

AD comes in two forms: familial early-onset AD, which is over 90% heritable and manifests prior to the age of 65 [24], and sporadic late-onset AD, which is more common, up to 78% heritable, and presents after the age of 65 [25]. The pathophysiology of sporadic AD is not as well understood and is likely due to a combination of genetic and environmental risk factors, but the pathophysiology of familial AD is more well-defined to specific causative genes and serves as a simpler starting point to exploring circuit-specific mechanisms. Our studies utilized a circuit-specific version of a rat model that recapitulates many of the features of familial AD in a manner that is translatable to relevant disease progression milestones seen in humans. Rats that globally express familial mutations in the hAPP+hPSEN1 genes accumulate Aβ plaques and display cognitive impairments beginning at 6 months of age, which corresponds to middle age in humans, and experience 40% degeneration of hippocampal neurons by 12 months of age, which corresponds to later adulthood [26,27]. This is consistent with humans, in whom Aβ plaques are thought to manifest as soon as 20 years prior to AD diagnosis [28], and the earlier soluble forms are thought to initiate disruptions to E/I ratio and network activity [29] that leads to cognitive dysfunction and comorbid symptoms such as psychosis in AD. Decreases in GABAergic interneurons positive for the calcium-binding protein PV, which synapse onto the cell body and axon hillock of glutamatergic pyramidal cell neurons they regulate, have been heavily implicated in the pathophysiology of both AD and psychosis [19], including ventral hippocampal disruptions to coordinated neuronal firing in a rat model used to study psychosis [20].

Here, in AAV-AD rats, we saw a 26% decrease in PV inhibitory interneurons, consistent with that seen in mice that globally express the same mutations [30] and neuronal and behavioral phenotypes consistent with comorbid psychosis in AD. Although our data do not display hippocampal hyperactivity as measured by firing rates of putative pyramidal cells using in vivo extracellular electrophysiology, this is consistent with the findings of a bigenic rat model used to study familial AD, in which patch clamp electrophysiology was instead able to measure other features consistent with hippocampal hyperactivity, such as increased resting membrane potential and wider action potential length in putative pyramidal cells [31]. Further, our data do demonstrate a decrease in vHipp oscillatory activity, which occurs at the population level in the context of hippocampal hyperactivity, specifically in the lower frequency delta bands (0–4 Hz) of hAPP + hPSEN1 rats compared to eGFP controls, which is restored by MP-III-022 compared to vehicle treatment.

These deficits in PV expression and spontaneous oscillatory activity in the vHipp likely drive downstream effects in the VTA and psychosis-like behaviors. AAV-AD rats displayed a nearly two-fold elevation in population activity and increased sensitivity to psychomotor stimulants in the locomotor activity assay, consistent with rodent models of psychosis [7,14,15,16,17] and sporadic AD [10], which were both reversed by administration of MP-III-022 compared to the vehicle. Interestingly, reduced PV neuron expression is likely to disinhibit pyramidal neurons due to reduced perisomatic and axon initial segment inhibition, whereas α5-PAM increases dendritic inhibition on the same neurons. The dendritic inhibition corresponds to SST input rather than PV input, thus representing a novel finding where deficits in PV function can be rescued through an SST-mediated cell mechanism (α5-PAM). Similar to our rodent model used to study sporadic AD [10], there were no significant differences in inhibition of the startle reflex; however, sensorimotor gating deficits can occur in numerous pathologies and thus may not always be as penetrant of a trait as increased sensitivity to psychomotor stimulants that are more specifically characteristic of psychosis in the clinic [32]. Additionally, these results were not surprising given that sensorimotor gating deficits involve brain regions that were not altered by pathology specific to the vHipp [33]. There were no significant differences in the percentage of spontaneous alternations in the Y-maze cognitive assay, which served to confirm the specificity of AD pathology specific to the vHipp in the model we used in this study, given deficits in short-term working memory would be expected with AD pathology in the dorsal hippocampus.

Since the hippocampus is a common site of primary pathology in both AD and psychosis, and previous work from our lab has shown hyperactivity of the hippocampus is both necessary and sufficient to induce neuronal and behavioral deficits analogous to psychosis in rats, these studies set out to establish that AD-like pathology specific to the vHipp is not only sufficient to model comorbid psychosis, but also necessary, given that drugs specific to the hippocampus may serve as a viable therapeutic strategy to target this pathophysiology in a circuit-specific manner. We found that viral expression of hAPP+hPSEN1 decreases oscillations and PV interneuron number in the vHipp, leading to increases in dopamine neuron population activity in the VTA and increased sensitivity to psychomotor stimulants. Furthermore, these deficits were reversed by the systemic administration of MP-III-022. These data mirror clinical and post-mortem brain studies [19,29], as well as preclinical work in other AD models [34]. Taken together, this work provides a circuit-based rationale for the use of AAV-AD rats as a translatable model that can be used to study drugs that target hippocampus-specific pathophysiology upstream of dopamine dysfunction that affects millions of patients worldwide who have no viable therapeutic options.

## 4. Materials and Methods

### 4.1. Animals

To generate a rodent model of AD (AAV-AD model; Figure 5), with pathological alterations restricted to the vHipp, we utilized high-titer adeno-associated viruses (AAV). Survival surgeries were performed in a semi-sterile environment under general anesthesia. Male (~325–350 g) Sprague-Dawley rats (Envigo, Indianapolis, IN, USA) were anesthetized with Fluriso™ (2–5% Isoflurane, USP with oxygen flow at 1 L/min) and placed in a stereotaxic apparatus (Kopf, Tujunga, CA, USA) using blunt atraumatic ear bars. A core body temperature of 37 °C was maintained. A mixture of two AAV9 containing vectors each expressing either the mutated form of hAPP (2.09 × 10^13^ GC/mL; pAAV[Exp]-CMV>{mutant hAPP}:WPRE) or hPSEN1 (6.28 × 10^13^ GC/mL; pAAV[Exp]-CMV>{hPSEN1}:T2A:TurboGFP:WPRE) containing a GFP reporter and driven by the CMV promotor were bilaterally injected (0.5 ul) into the vHipp (A/P: −5.3 mm and M/L: ±5.0 mm from Bregma; D/V: −9.0 mm ventral of the brain surface). Control rats were bilaterally injected with an AAV expressing eGFP (2.09 × 1013 GC/mL; pAAV[Exp]-CMV>EGFP:WPRE). Rats received post-operative ketoprofen (5 mg/kg; s.c.) and were housed in groups of 2–3 until ~8 months of age prior to behavioral testing and electrophysiological recordings. An experimental timeline is depicted in Figure 1A. A subset of control and AAV-AD rats were treated systemically with either vehicle (1% Tween 80, 14% propylene glycol, in distilled water) or the selective α5-PAM, MP-III-022 (10 mg/kg; i.p.), 20 min prior to any electrophysiological or behavioral testing. The dose was chosen based on previously published literature using this compound [11,35,36].

### 4.2. In Vivo Extracellular Electrophysiology Recordings

For non-survival surgery, rats were anesthetized with chloral hydrate (400 mg/kg; i.p.) prior to placement in a stereotaxic apparatus (Kopf; Tujunga, CA, USA). This anesthetic is required for dopamine neuron physiology, as it does not significantly alter dopamine activity when compared to recordings in freely moving animals [37]. A core body temperature of 37 °C was maintained, and supplemental anesthesia was administered as required to maintain suppression of the limb withdrawal reflex. Extracellular glass microelectrodes (impedance 6–10 MΩ) were lowered into the vHipp (A/P: −5.3 and M/L: ±5.0 mm from Bregma; D/V: −4.5 to −9.0 mm ventral of the brain surface) or VTA (A/P: −5.3 mm and M/L: ±0.6 mm from Bregma; D/V: −6.5 to −9.0 mm ventral of the brain surface). The firing frequency of spontaneously active putative pyramidal neurons in the vHipp was measured and identified as previously published (neurons with firing frequencies less than 2 Hz) [12,13,38]. Local field potential (LFP) oscillatory activity was generated from the vHipp using open filter settings (Low Pass Filter: 0.3 Hz; High Pass Filter: 1000 Hz) and recorded for a minimum of ten minutes. Oscillations were quantified with commercially available computer software (LabChart version 8; ADInstruments, Chalgrove, Oxfordshire, UK). Spontaneously active dopamine neurons were identified using previously established criteria [39,40]: (1) action potential duration > 2 ms and (2) frequency between 0.5 and 15 Hz. Three parameters of dopamine activity were measured: (1) population activity (the number of spontaneously active dopamine neurons encountered per track); (2) basal firing rate; (3) and the proportion of action potentials occurring in bursts (defined as the incidence of spikes with <80 ms between them; termination of the burst is defined by >160 ms between spikes). Recordings typically lasted for ~2–4 h. Electrophysiological recordings were analyzed by two-way ANOVA (strain × frequency) followed by a Holm-Sidak post-hoc test.

### 4.3. Y-Maze Spontaneous Alternation Assay

The Y-maze consists of three plastic arms (81 cm long × 20.32 cm wide × 20 cm high) separated by 120-degree angles. Rats were placed inside the same arm facing the center of the maze and allowed to move freely for 10 min. During this time, the number and order of arm entries (defined as at least 80% of the front of the rat entering an arm from the center of the maze) were recorded. To calculate percent alternations, the number of alternations (triad containing entry to all three arms) was divided by the total number of entries minus 2 and multiplied by 100. Y-maze data were analyzed by two-way ANOVA followed by a Holm-Sidak post hoc test.

### 4.4. Pre-Pulse Inhibition of Startle (PPI)

Rats were placed in a sound-attenuated chamber (SD Instruments; San Diego, CA, USA) and allowed to acclimate to 65 dB background noise for 5 min prior to exposure to 10 startle-only trials [40 ms, 120 dB, 15 s average inter-trial intervals (ITIs)], followed by 24 trials where a pre-pulse (20 ms at 69 dB, 73 dB, and 81 dB) was presented 100 ms prior to a startle pulse. Each pre-pulse was presented six times in a pseudo-random order (15 s ITI). Startle responses were measured from 10 to 80 ms after the onset of the startle-only pulse and recorded using SR-Lab Analysis Software (https://sandiegoinstruments.com/product/sr-lab-startle-response/ accessed on 5 May 2022). PPI data were analyzed by three-way ANOVA (strain × treatment × dB) followed by a Holm-Sidak post hoc test.

### 4.5. Stimulant-Induced Locomotor Activity 

Rats were placed in an open field arena (Med Associates, St. Albans, VT, USA) and spontaneous locomotor activity in the x-y plane was determined by beam breaks and recorded with Open Field Activity software (version 5). After 30 min of baseline recording, all rats were injected with the NMDA receptor antagonist, MK-801 (0.075 mg/kg, i.p.), and recorded for an additional 45 min. Locomotor data were analyzed by two separate (baseline and following MK-801) three-way ANOVAs (strain × treatment × time) followed by a Holm-Sidak post hoc test.

### 4.6. Immunohistochemistry

At the cessation of all experiments, rats were transcardially perfused with saline (150 mL), and brains were extracted and post-fixed for at least 24 h (4% formaldehyde in 0.1 M phosphate buffered saline (PBS) prior to cryoprotecting in 10% sucrose (dissolved in PBS), and coronally sectioned (50 µm) using a cryostat (Leica, Buffalo Grove, IL, USA). A subset of vHipp slices was mounted and cover slipped with Prolong™ Gold anti-fade mountant to verify viral expression (by GFP fluorescent reporter; Figure 2B,C). To verify the presence of amyloid plaques in the vHipp, a subset of slices containing the vHipp were silver stained using the FD NeuroSilver Kit II according to the manufacturer’s directions, and cover slipped with Permount^®^ (Figure 1C). A separate subset of slices containing the vHipp was used to detect the expression of PV (Figure 5B). In short, slices were washed three times (10 min each) in PBS, blocked (2% normal goat serum and 0.3% Triton™ X 100 in PBS) for 30 min at room temperature, and incubated with rabbit anti-parvalbumin primary antibody (1:1000) overnight at 4 °C. After washing (three times in PBS for 10 min each), slices were incubated with AlexaFluor^®^ 594 goat anti-rabbit immunoglobin G (H + L) (1:1000) for 1 h at room temperature. Slices were wet mounted, and cover slipped with ProLong™ Gold anti-fade mountant. Sections were imaged using an AxioCam ICc 1 (Zeiss, Jena, Germany) camera attached to an Axio Lab.A1 (Zeiss, Jena, Germany) microscope. The number of PV-positive cells within the vHipp was counted on six sections per animal, in an area approximately −6.0 to −8.5 mm ventral of the skull surface and ±4.0 to 6.0 mm lateral of the midline.

### 4.7. Histology

A subset of brains was post-fixed (4% formaldehyde in PBS), cryoprotected (10% *w*/*v* sucrose in PBS) until saturated, and coronally sectioned (25 µm) on a cryostat (Leica, Buffalo Grove, IL, USA). Sections were mounted onto gelatin-chrome-coated slides, stained with neutral red (0.1%) and thionin acetate (0.01%), and cover slipped with DPX mountant for histological verification of electrode tracks within the vHipp (Figure 2A) or VTA (Figure 3A) [41].

### 4.8. Analysis

Electrophysiological analysis of vHipp and dopamine neuron activity was performed with commercially available computer software (LabChart version 8; ADInstruments, Dunedin, New Zealand; Chalgrove, Oxfordshire, UK). Locomotor activity was collected with Activity Monitor software (version 5; MED Associates; St. Albans, VT, USA). PPI data were collected using SR-Lab™ Analysis Software (https://sandiegoinstruments.com/product/sr-lab-startle-response/ accessed on 5 May 2022) (SD Instruments; San Diego, CA, USA). All data were analyzed using Prism software (version 10; GraphPad Software Inc.; San Diego, CA, USA). Data are represented as mean ± SEM with n values representing the number of animals per experimental group unless otherwise stated. Data were analyzed using a *t*-test, linear regression, or two- or three-way analysis of variance (ANOVA), and the Holm-Sidak post hoc test was used when significant interactions were determined. Statistics were calculated using SigmaPlot (version 12; Systat Software Inc.; Chicago, IL, USA) and significance was determined at *p* < 0.05.

### 4.9. Materials

Proprietary compound, MP-III-022, was generated by the University of Wisconsin in Milwaukee. Plasmids were generated and packaged into AAV by VectorBuilder (Chicago, IL, USA). FlurisoTM was purchased from MWI Animal Health (Boise, ID, USA). Chloral hydrate (C8383), Ketoprofen (K2012), MK-801 (M107), Propylene Glycol (P4347), Tween^®^ 80 (P1754), and DPX mountant (06522) were sourced from Sigma-Aldrich (St. Louis, MO, USA). Anti-parvalbumin antibody (ab11427) was purchased from Abcam (Boston, MA, USA). AlexaFluor^®^ 594 goat anti-rabbit immunoglobin G (H + L) (A-11012) was purchased from Invitrogen (Waltham, MA, USA). Invitrogen™ ProLong™ Gold antifade mountant (P36930) and Permount™ mounting medium (SP15) were purchased from Thermo Fisher Scientific (Waltham, MA, USA). FD NeuroSilver Kit II (PK301) was sourced from FD NeuroTechnologies, Inc. (Columbia, MD, USA). All other chemicals and reagents were either analytical or laboratory-grade and purchased from standard suppliers. Interventional studies involving animals or humans and other studies that require ethical approval must list the authority that provided approval and the corresponding ethical approval code.

## Figures and Tables

**Figure 1 ijms-24-11788-f001:**
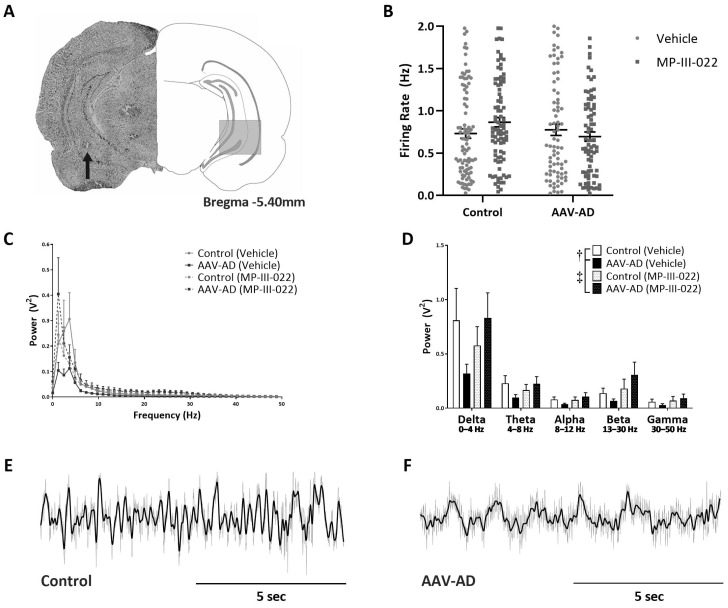
Impaired ventral hippocampal oscillatory activity reversed by MP−III−022 in AAV−AD rats. Representative brain slice with an electrode track (indicated by black arrow) in the ventral hippocampus ((**A**), left) and corresponding schematic of the brain section ((**A**), right). The average firing rate of spontaneously active putative pyramidal neurons in the ventral hippocampus (vHipp) is unchanged between treatment groups and unaffected by treatment with the selective α5−GABAA PAM, MP−III−022 (**B**). AAV−AD rats display a significant decrease in oscillatory power when compared to eGFP controls, which was reversed by the systemic administration of MP−III−022 (**C**,**D**). Representative trace of spontaneous local field potential oscillations filtered for delta (dark line) in the vHipp of a control (**E**) and AAV−AD rat (**F**). † denotes a significant main effect; *p* = 0.014. ‡ denotes a significant main effect; *p* = 0.006.

**Figure 2 ijms-24-11788-f002:**
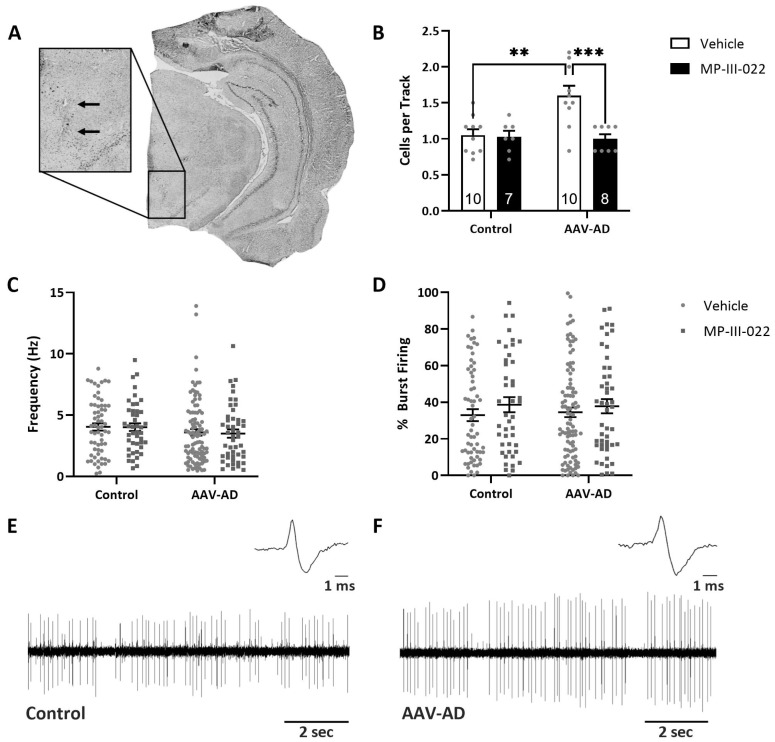
Aberrant ventral tegmental area activity is restored by MP-III-022 in AAV-AD rats. Representative brain slice with an electrode track (indicated by black arrows) in the ventral tegmental area (**A**). Dopamine neuron population activity (average number of spontaneously active dopamine neurons per electrode track) is significantly increased in AAV-AD rats, which is reversed by systemic administration of the selective α5-GABAA PAM, MP-III-022 (**B**). The average firing rate (**C**) or bursting pattern (**D**) was not altered in control or AAV-AD rats or by systemic MP-III-022 administration. Representative dopamine recording and action potential from a control (**E**) and AAV-AD (**F**) rat. ** *p* < 0.005, *** *p* < 0.0001.

**Figure 3 ijms-24-11788-f003:**
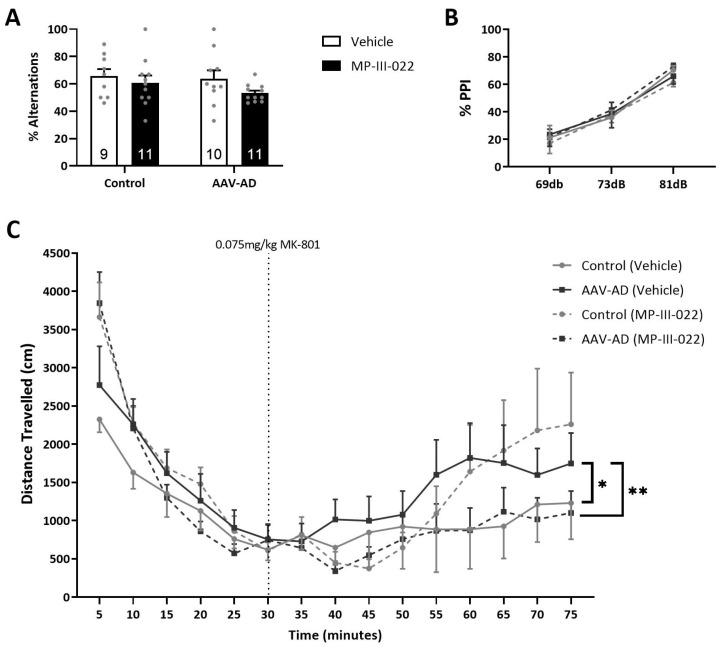
AAV-AD rats exhibit an enhanced sensitivity to the locomotor-inducing effects of MK-801, which are reversed by MP-II-022. No significant working memory deficits were observed between any groups in Y-maze (**A**). Additionally, no significant sensorimotor gating differences were observed, as measured by pre-pulse inhibition of startle (**B**). AAV-AD rats displayed an enhanced locomotor response to MK-801, which was reversed by the systemic administration of the selective α5-GABAA positive allosteric modulator, MP-III-022 (**C**). * *p* = 0.026, ** *p* = 0.004.

**Figure 4 ijms-24-11788-f004:**
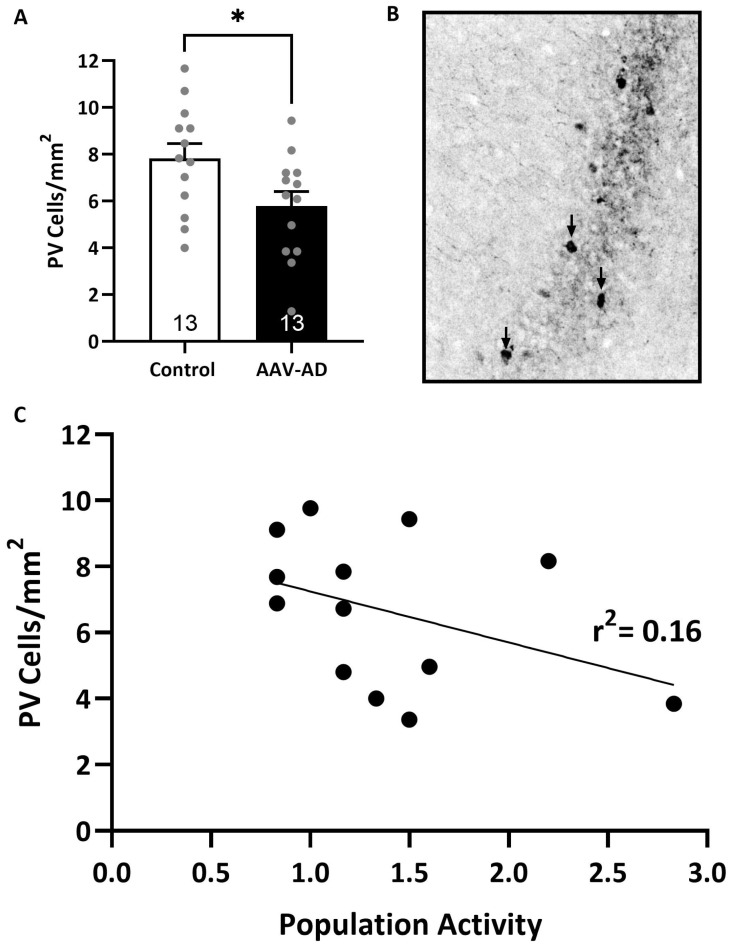
AAV-AD rats exhibit a significant decrease in the number of parvalbumin-positive (PV) interneurons in the vHipp. AAV-AD rats exhibit significantly less parvalbumin-positive inhibitory interneurons in the vHipp (**A**). Representative coronal brain section containing the ventral hippocampus (vHipp) displaying parvalbumin (converted to greyscale and indicated by black arrows; (**B**)). Scatterplot depicting correlation between dopamine neuron population activity and the number of PV interneurons in the vHipp. Black dots represent individual rats. (**C**). * *p* = 0.032.

**Figure 5 ijms-24-11788-f005:**
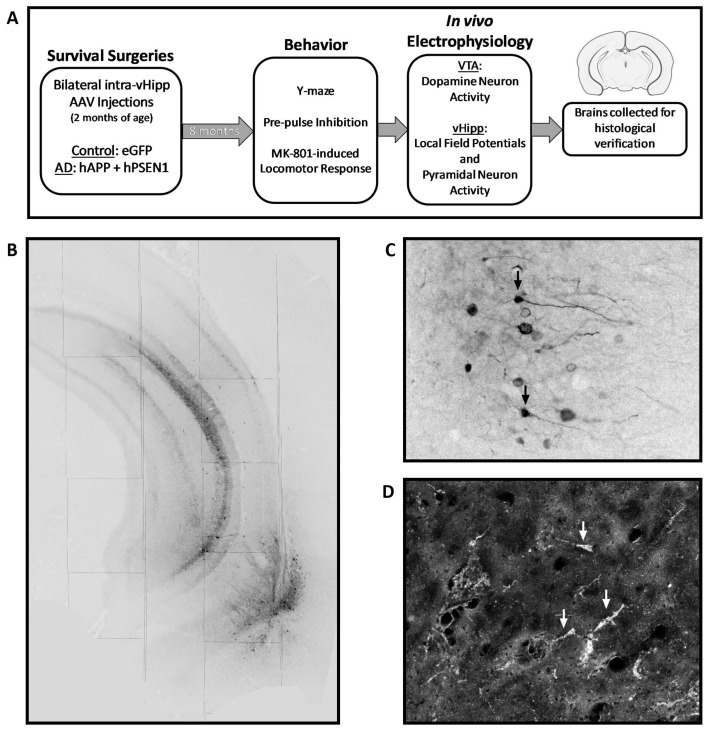
Verification of the AAV-AD model. Schematic representation of the experimental timeline (**A**). Representative coronal brain slice verifying viral reporter expression within the ventral hippocampus. Black arrows indicate representative neurons (converted to greyscale (**B**,**C**)). Representative dark field image of a silver-stained ventral hippocampal coronal brain slice, with pathological deposits in the vHipp of AAV-AD rats, indicated by the white arrows in (**D**). Icons used were adapted from BioRender.com (2022). Retrieved from http://app.biorender.com/illustrations (accessed on 5 May 2022).

## Data Availability

Data available on request.

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
