# Peer review of "α5-GABAA Receptor Modulation Reverses Behavioral and Neurophysiological Correlates of Psychosis in Rats with Ventral Hippocampal Alzheimer’s Disease-like Pathology"

_ijms, 2023, doi:10.3390/ijms241411788_

Round 1

Reviewer 1 Report

The manuscript entitled “α5-GABAA receptor modulation reverses behavioral and neurophysiological correlates of psychosis in rats with ventral hippocampal Alzheimer’s Disease-like pathology”, Alzheimer’s Disease (AD) is the most common neurodegenerative disease. 35 million people in the world suffering with Alzheimer’s Disease (AD), up to half experience comorbid psychosis. Antipsychotics, used to treat psychosis, are contraindicated in elderly patients because they increase the risk of premature death. Reports indicate that the hippocampus is hyperactive in patients with psychosis and those with AD. The manuscript provides the evidence for the development of drugs that target 5-GABAA receptors for patients with AD and comorbid psychosis. The manuscript is well organized. It would be reasonable to accept in IJMS.

Author Response

We thank the reviewer for their careful review of our manuscript.

Reviewer 2 Report

This article investigates the association of psychosis and Alzheimer’s Disease (AD) in the Ventral Hippocampus (vHipp) by using α5-GABAA receptor modulator drug. Authors have generated an AVV-Alzheimer rat model by virally expressing known human mutations in human amyloid precursor protein (hAPP) and presenilin1 (hPSEN1) specifically in the vHipp. These rats were then used for neuronal pathology, behavioral and cognitive tests. A significant increase in dopamine neuron population activity was observed in these AD-AAV models. Systemic administration of positive allosteric modulators of α5 (α5-PAMs), MP-III-022 in vHipp reversed the dopamine-associated changes observed in AAV-AD rats. These kinds of studies are very important to find circuits and potential new therapeutic targets for AD and its associated comorbidities. This article is written very clearly with proper literature and explanation. However, the authors need to make some minor changes and better proofreading to improve the quality of this paper.

-          Figure 1: 1C and ID, x- and y-axis labels are too small to read.

-          Line 247: AAV-AD model is shown in Figure 5 and not in “Figure 1” as written in line 247.

-          The authors need to clarify the titer of the AAV vectors. e.g. Is 2.09x1013 GC/ml the right concentration of the vector used?

-          Line 24, 26 in the abstract:  α symbol is not proper

-          Line 348: temperature symbol is not proper

-          Line 26, 124, 295, 348: too much spacing

-          In vivo should be in italics

English is quite good. Only some minor corrections are required.

Author Response

  • “…the authors need to make some minor changes and better proofreading to improve the quality of this paper.” We appreciate the reviewers careful review of this manuscript and we have proofread the text for spelling and grammar.
  • “Figure 1: 1C and ID, x- and y-axis labels are too small to read.” The size of x- and y-axis labels have been increased for better visibility.
  • “Line 247: AAV-AD model is shown in Figure 5 and not in “Figure 1” as written in line 247.” The figure number has been corrected.
  • “The authors need to clarify the titer of the AAV vectors. e.g. Is 2.09x1013 GC/ml the right concentration of the vector used?” We apologize for the typo and concentrations have been corrected.
  • “Line 24, 26 in the abstract:  α symbol is not proper. Line 348: temperature symbol is not proper” The symbols have been updated.
  • “Line 26, 124, 295, 348: too much spacing” Excess spacing has been corrected on all lines.
  • In vivo should be in italics” In vivo has now been italicized.